# Efficacy and Safety of Pembrolizumab for Gemcitabine/Cisplatin-Refractory Biliary Tract Cancer: A Multicenter Retrospective Study

**DOI:** 10.3390/jcm9061769

**Published:** 2020-06-07

**Authors:** Sang Hoon Lee, Hee Seung Lee, Sang Hyub Lee, Sang Myung Woo, Dong Uk Kim, Seungmin Bang

**Affiliations:** 1Department of Internal medicine, Severance Hospital, Yonsei University College of Medicine, Seoul 03772, Korea; lshjjang_2000@yuhs.ac (S.H.L.); lhs6865@yuhs.ac (H.S.L.); 2Pancreaticobiliary Cancer Study Group of Korean Society of Gastrointestinal Cancer, Seoul 03741, Korea; gidoctor@snuh.org (S.H.L.); wsm@ncc.re.kr (S.M.W.); amlm3@hanmail.net (D.U.K.); 3 Digestive Disease Center, Department of Internal Medicine, Konkuk University Medical Center, Konkuk University School of Medicine, Seoul 05030, Korea; 4Department of Internal medicine and Liver Research Institute, Seoul National University Hospital, Seoul National University College of Medicine, Seoul 03080, Korea; 5Center for Liver and Pancreaticobiliary Cancer, National Cancer Center, Goyang 10408, Korea; 6Biomedical Institute, Pusan National University Hospital, Busan 49241, Korea

**Keywords:** biliary tract neoplasm, cholangiocarcinoma, immunotherapy, PD-L1 costimulatory protein, biomarker

## Abstract

Pembrolizumab, an anti-programmed cell death (PD)-1 monoclonal antibody, is an anticancer agent showing substantial benefit in lung cancer and melanoma treatment. Biliary tract cancer (BTC) has been shown to respond to pembrolizumab; however, no credible data of such treatment outcomes exist. Therefore, we assessed the clinical outcomes and safety of pembrolizumab in patients with gemcitabine/cisplatin-refractory BTC. In this multicenter study, we retrospectively analyzed 51 patients with programmed cell death 1-ligand 1 (PD-L1)-positive gemcitabine/cisplatin-refractory BTC treated with pembrolizumab in four tertiary hospitals in Korea. PD-L1 positivity was defined as the expression of PD-L1 in ≥1% of tumor cells based on immunohistochemical staining (22C3, SP263, and E1L3N assays). The median age of the patients was 66 (range, 43–83) years and 29 (56.9%) were male. Extrahepatic cholangiocarcinoma was the most common cancer type (*n* = 30, 58.8%). Partial response and stable disease were achieved in 5 (9.8%) and 13 (25.5%) patients, respectively. Median progression-free survival and overall survival were 2.1 (95% CI, 1.7–2.4) and 6.9 (95% CI, 5.4–8.3) months, respectively. Overall, 30 (58.8%) patients experienced treatment-related adverse events (AEs). Only four (7.8%) patients experienced grades 3 and 4 AEs. In PD-L1-positive gemcitabine/cisplatin-refractory BTC, pembrolizumab presented durable efficacy, with a 9.8% response rate and manageable toxicity.

## 1. Introduction

Biliary tract cancers (BTCs) consist of all tumors originating in the epithelial cells of the bile duct and they are classified into intrahepatic cholangiocarcinoma (CCA), extrahepatic CCA (including perihilar and distal CCA), and gallbladder cancer, according to their anatomical location within the biliary tree. More than 90% of BTCs are adenocarcinoma. Although the incidence of BTC is the highest in East and South Asia and parts of South America, its global incidence has been increasing significantly [1,2]. Surgical resection is regarded as the curative treatment option for patients with BTC but most patients are not considered suitable candidates for surgery because of an advanced disease status at diagnosis [3,4]. Patients with advanced BTC have a poor prognosis with a dismal survival time of <1 year [5,6]. Recent updated guidelines recommend gemcitabine plus cisplatin as the first-line chemotherapy for patients with advanced BTC [4,7]. However, effective treatment options for patients with gemcitabine/cisplatin-refractory BTC are currently limited [8].

Pembrolizumab, an anti-programmed cell death (PD)-1 monoclonal antibody, is an anticancer agent that has shown substantial benefit in patients with lung cancer and melanoma [9,10,11]. Following the discovery of the PD-1/PD-1-ligand 1 (PD-L1) pathway as an immunological target to control cancer, immune checkpoint inhibitors, including pembrolizumab, have been designed to overcome the inhibitory responses of cytotoxic T lymphocytes to promote antitumor immune responses [12,13,14]. The first related study showed a notable clinical effect of pembrolizumab in patients with non-colorectal cancer, including four with BTC with mismatch repair (MMR) deficiency, with a 71% objective response rate (ORR) [13]. In a few subsequent studies, pembrolizumab was reported to be effective against advanced BTC, targeting a subpopulation with more than 1% PD-L1 expression in tumors [15,16]. However, there are no credible data of the treatment outcomes of pembrolizumab against BTC. In the present multicenter retrospective study, we investigated the clinical outcomes and safety profile of pembrolizumab in patients with gemcitabine/cisplatin-refractory BTC using PD-L1 positivity as a response biomarker and attempted to identify a subpopulation that would potentially benefit from this treatment.

## 2. Patients and Methods

### 2.1. Study Population

From August 2017 to April 2019, 65 patients with PD-L1-positive gemcitabine/cisplatin-refractory BTC who received pembrolizumab were included in this multicenter study with the following inclusion criteria: (1) ≥20 years of age; (2) histologically or cytologically confirmed locally advanced or metastatic BTC; (3) at least one measurable or evaluable lesion according to Response Evaluation Criteria in Solid Tumors (RECIST) version 1.1 [17]; (4) radiologically confirmed presence of progressive disease or intolerance to gemcitabine/cisplatin chemotherapy; (5) ≥1% PD-L1-positive tumor cells as assessed using immunohistochemical (IHC) staining; and (6) receipt of at least one cycle of pembrolizumab.

Fourteen patients were excluded according to the following criteria: (1) diagnosis of ampulla of Vater cancer (*n* = 1); (2) inability to assess tumor response (*n* = 6); (3) no gemcitabine/cisplatin chemotherapy before pembrolizumab injection (*n* = 4); and (4) no PD-L1 IHC-staining examination (*n* = 3). Fifty-one patients (41 from Severance Hospital, Seoul, Korea; 4 each from Seoul National University Hospital, Seoul, Korea, and National Cancer Center, Goyang, Korea; and 2 from Pusan National University Hospital, Busan, Korea) were included in the final analysis. Among them, 44 patients (86.3%) were radiologically confirmed to have progressive disease, and another 7 (13.7%) were intolerant to gemcitabine/cisplatin chemotherapy. All the clinical, laboratory and radiologic data were collected from electronic medical records and were retrospectively reviewed without obtaining informed consent. The study was approved by the institutional review board of each institution (Severance Hospital Institutional Review Board, Seoul National University Hospital Institutional Review Board, National Cancer Center Institutional Review Board, and Institutional Review Board of Pusan National University Hospital).

### 2.2. PD-L1 IHC Assay

PD-L1 expression was assessed by conducting an IHC staining of achieved tumor tissues before systemic treatment, using the E1L3N (Cell Signaling Technology, Danvers, MA, USA), 22C3 (Agilent Technologies, Santa Clara, CA, USA), and SP263 (Ventana Benchmark Ultra, Tuscon, AZ, USA) assays. PD-L1-positive tumor cells were considered if the viable tumor cells exhibited any perceptible, partial or complete, membranous or cytoplasmic staining, as previously described [18]. PD-L1-positive status was defined based on a 1% threshold in immunostained tumor cells in the entire tumor section by any IHC method. The frequency of use of the E1L3N, 22C3, and SP263 IHC assays was as follows: *n* = 33 (57.9%), 27 (47.4%), and 9 (15.8%), respectively. PD-L1 expression was categorized into three subgroups based on the proportion of immunostained tumor cells using additional cutoff values of 5% and 50%.

### 2.3. Treatment Schedule and Response Evaluation

All patients received 200 mg pembrolizumab intravenously, every 3 weeks after a 17-day median interval from the last prior treatment. Dose reduction, administration delay, or both were performed if serious treatment-related adverse events (AEs) developed, making treatment intolerable. Pembrolizumab administration was interrupted when disease progression or life-threatening AEs were identified. To evaluate treatment efficacy, we routinely evaluated the tumor response every three cycles using abdominal or chest computed tomography or both according to RECIST version 1.1 [17].

### 2.4. Assessment of Treatment-Related AEs

To monitor treatment-related AEs, physicians and registered nurses meticulously evaluated the occurrence of AEs at each visit during treatment. The category and severity grade of the AEs were accurately recorded in the medical records of the patients. Treatment-related AEs were evaluated according to National Cancer Institute Common Toxicity Criteria version 4.0. Treatment delays or discontinuations associated with the AEs were also documented with the reason.

### 2.5. Study Endpoints and Statistical Analysis

The primary endpoint of this study was response rate and the secondary endpoints were AEs, progression-free survival (PFS), and overall survival (OS). Tumor responses included complete response (CR), partial response (PR), stable disease (SD), and progressive disease (PD). DCR was defined as the summation of CR, PR, and SD. PFS was defined as the time from the initiation of pembrolizumab treatment to disease progression or date of death, and OS was defined as the time from the initiation of pembrolizumab treatment to the date of death due to any cause or the last follow-up visit. The last follow-up date was 31 September 2019, and the median follow-up duration was 3.8 (range, 0.6–18.4) months.

Data are expressed as the mean ± standard deviation, median (range), or *n* (%), as appropriate. A survival analysis for PFS and OS was performed using the Kaplan-Meier method and compared using the log-rank test. To identify independent risk factors for progression, we performed a multivariate Cox proportional hazard regression analysis using the significant variables in the univariate analysis. Hazard ratios (HRs) and the corresponding 95% confidence intervals (CI) were also calculated. A two-tailed *p*-value of 0.05 was considered statistically significant, and the statistical analysis was performed using SPSS version 25.0 (PASW Statistics Inc., Chicago, IL, USA).

## 3. Results

### 3.1. Baseline Characteristics

The baseline characteristics of the 51 enrolled patients are presented in Table 1. The median age was 66 (range, 43–83) years and 29 (56.9%) patients were male. Extrahepatic CCA was the most common type of cancer (*n* = 30, 58.8%), followed by intrahepatic CCA (*n* = 12, 23.5%) and gallbladder cancer (*n* = 9, 17.6%). All patients were diagnosed with adenocarcinoma, histologically or cytologically. Most patients were diagnosed at a metastatic stage (*n* = 45, 88.2%), including 28 with post-operative recurrence. Common organs of distant metastasis were as follows: liver (*n* = 24, 47.1%), intra-abdominal lymph node (*n* = 24, 47.1%), peritoneal seeding (*n* = 17, 33.3%), bone (*n* = 4, 7.8%), and lungs (*n* = 2, 3.9%). Histological grading was moderately and poorly differentiated in 28 (54.9%) and 14 (27.5%) patients, respectively. Subgrouping based on the proportion of PD-L1-positive tumor cells with cutoff values of 5% and 50% revealed that low (1–5%), moderate (5–50%), and high (≥50%) PD-L1 subgroups accounted for 35 (68.6%), 11 (21.6%), and 5 (9.8%) patients, respectively. The median carbohydrate antigen (CA) 19-9 level was 276.6 U/mL (range, 1.1–24,253.0). Pembrolizumab was administered as the second, third, and fourth or greater line chemotherapy in 33 (64.7%), 14 (27.5%), and 4 (7.8%) patients, respectively. A median of three cycles (range 1–15) of pembrolizumab was administered during a median of 3.8 (range, 0.6–18.4) months of follow-up.

### 3.2. Clinical Outcomes and AEs

In the 51 patients enrolled, PR and SD were achieved in 5 (9.8%) and 14 (25.5%) patients, respectively (Table 2), whereas the DCR was 35.2%. In five patients with PR, the median time to response was 2.5 (range, 2.1–8.3) months and the median duration of treatment was 9.1 months. Five patients with PR exhibited 1, 2, 5, 10, and 50% of PD-L1 expression level. Overall, the median PFS and OS were 2.1 (95% CI, 1.7–2.4) and 6.9 (95% CI, 5.4–8.3) months, respectively (Figure 1). Figure 2 illustrates the changes in tumor size from the baseline to the best response in 42 patients. When the study population was divided into three subgroups based on PD-L1 expression, this did not appear to be significantly associated with the best tumor response (Figure 2) 

Overall, 30 (58.8%) patients experienced treatment-related AEs, with fatigue (*n* = 11, 21.6%, Table 3) being the most common. Four (7.8%) patients experienced grade 3 and 4 AEs (consisting of grade 3 nausea, fatigue, poor oral intake, diarrhea, and drug-induced pneumonitis) and two patients (3.9%) discontinued pembrolizumab treatment owing to AEs. There was no case of treatment-related death.

### 3.3. Factors Associated with Progression

Univariate analysis showed that the number of prior therapies (≥2), hemoglobin level (<10 g/dL), and CA 19-9 level (>500 U/mL) were significantly associated with progression (Table 4). Subgroups based on PD-L1 expression could not be used to predict progression after pembrolizumab treatment. Although the subsequent multivariate analysis did not identify statistically significant predictors of progression, patients receiving pembrolizumab as second-line chemotherapy with a normal hemoglobin level tended to exhibit favorable PFS (*p* = 0.069 and *p* = 0.062, respectively).

## 4. Discussion

In this multicenter study, pembrolizumab showed durable antitumor activity, with a 9.8% response rate and manageable toxicity in patients with PD-L1-positive gemcitabine/cisplatin-refractory BTCs. The first BTC cohort from the KEYNOTE-028 open-label, phase I clinical trial, which included patients with 20 different types of solid cancers with PD-L1 expression of >1% (determined by IHC-based assay), revealed an ORR of 17% (one CR and three PR) and a DCR of 34% (four additional SD) among 23 patients with BTC [15,19]. The results of another prospective cohort study of 40 patients with BTC with similar characteristics in a single medical institution showed an OCR of 10.0% (four patients with PR) and a DCR of 47.5% (additional 15 patients with SD) [16]. Our study had a larger population scale than these two studies but showed a similar overall response rate (Table 5). Intriguingly, some patients who responded showed a long duration of treatment: >6 months (median 9.1 months in our study; 6.3 months [16] and 2-year duration of treatment of 67% [15] and 50% [20] patients, respectively, in other studies).

The interim report of KEYNOTE-158, the largest ongoing phase II, single-arm, open-label cohort study of pembrolizumab in patients with various types of advanced cancers, including 104 patients with BTC, was published in 2018 [20]. The 104 patients with BTC showed an ORR of only 5.8%, including 0, 6, and 17 patients with CR, PR, and SD, respectively (DCR was 22.1%) [20]. This cohort study included 61 and 31 patients with PD-L1-positive and PD-L1-negative expression, respectively. Interestingly, PD-L1 expression did not predict treatment response and this is similar to the findings of our study. Although the ORR was slightly higher in the PD-L1-positive group than in the negative group (6.6% vs. 2.9%), there were no significant differences in the median PFS (1.9 vs. 2.1 months) or OS (7.2 vs. 9.6 months) [20]. This result, which showed the limited clinical efficacy of pembrolizumab monotherapy in advanced BTC, encouraged us to explore biomarkers to identify a subpopulation likely to respond and a new therapeutic strategy to overcome the limited antitumor response in BTC.

The IHC analysis of PD-L1 expression predicts the clinical efficacy of immune checkpoint inhibitors in several cancers [16,21,22,23]. In particular, in non-small-cell lung cancer, the cutoff value of PD-L1 was lowered from 50% to 1%, and immune checkpoint inhibitors are now considered as the first-line treatment for advanced stage tumor with or without conventional chemotherapy [21,23,24]. Recently, Kang et al. suggested that a high cutoff value of ≥50% could predict tumor response when evaluated by immune-modified RECIST criteria [16]. In addition, a Japanese phase I study of another PD-L1 inhibitor, nivolumab, reported that PD-L1-positive patients (≥1%) had longer median PFS (2.8 vs. 1.4 months) and OS (11.6 vs. 5.2 months) than those of PD-L1-negative patients among 30 patients with gemcitabine-refractory advanced BTC [25]. These findings, however, resulted from preliminary studies with relatively small-sized study populations and did not correspond with the results of KEYNOTE-158 or our study [20]. Furthermore, our analysis of several clinical factors to validate another response predictor of pembrolizumab revealed few prior therapies and normal hemoglobin level as predictors of a favorable tendency to respond, without statistical significance in the multivariate analysis. Hence, the usefulness of these factors, including tumor PD-L1 status, should be verified in large-scale, prospective clinical trials in the future.

An established biomarker for patients’ response to immune checkpoint inhibitors is the neo-antigen tumor burden resulting from microsatellite instability (MSI) owing to MMR deficiency (MSI-high/MMR-deficiency) [26]. The genomic spectra of BTC have shown the presence of MSI in a small portion of approximately 1–2% [26,27,28,29]. The clinical results of pembrolizumab monotherapy in advanced MSI-high/MMR-deficient cancers, including 4 and 11 patients with BTC, demonstrated ORR values of 25% and 27%, respectively, for a subset of patients with BTC with a response duration of >6 months [13,30]. However, we could retrospectively assess the MSI status in seven (13.7%) patients in our study population, none with MSI-H/MMR-deficiency.

To evaluate the significant response biomarker, the KEYNOTE-028 (NCT02054806) study analyzed multiple biomarkers including expression of an 18-gene T cell-inflamed profile, PD-L1 expression, and tumor mutational burden. All three factors (alone or combined) were correlated with high response rate to pembrolizumab in the overall study population of multiple tumor types [19]. However, biomarker data specific for BTC has not been indubitably presented until now.

The safety profile of pembrolizumab in this study was generally consistent with that in previous studies [9,13,15,16]. Four (7.8%) patients experienced grade 3 AEs with 50% (*n* = 2, 3.9%) treatment discontinuation in our study. Two patients with pneumonitis associated with immune checkpoint inhibition were conservatively managed with one week of hospitalization. There are some limitations in our study. Although all patients in the study were enrolled in a number of tertiary care hospitals, the possibility of selection bias cannot be excluded due to its retrospective nature. The relatively small study population may also be insufficient to perform a statistically robust analysis. Additionally, inconsistent IHC assay methods could serve as a limitation for further investigation.

In conclusion, pembrolizumab monotherapy in patients with gemcitabine/cisplatin-refractory BTC and PD-L1 positivity showed long-lasting anticancer effects in approximately 10% of the study population with an overall manageable safety profile. Intensity of PD-L1 expression was not supposed to an effective surrogate marker for the efficacy of pembrolizumab; therefore, further investigation is needed to confirm an appropriate biomarker for selecting the subpopulation of advanced BTC patients who might be sensitive to immune checkpoint inhibitors.

## Figures and Tables

**Figure 1 jcm-09-01769-f001:**
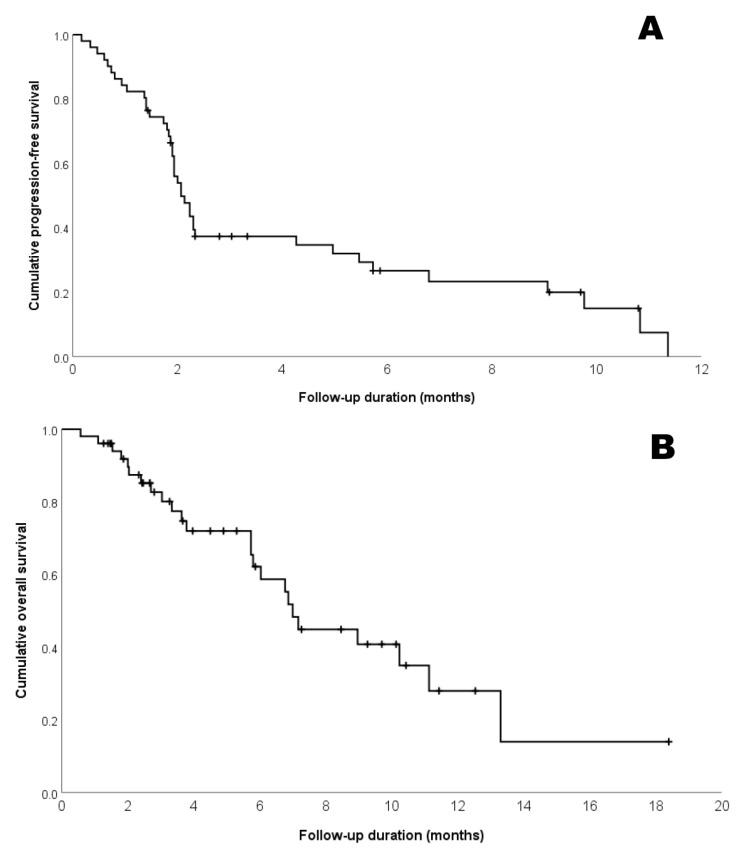
Kaplan-Meier estimates of the (**A**) progression-free survival (PFS); and (**B**) overall survival (OS) of the entire study population (*n* = 51). Median PFS and OS were 2.1 (95% confidence interval (CI), 1.7–2.4) and 6.9 (95% CI, 5.4–8.3) months.

**Figure 2 jcm-09-01769-f002:**
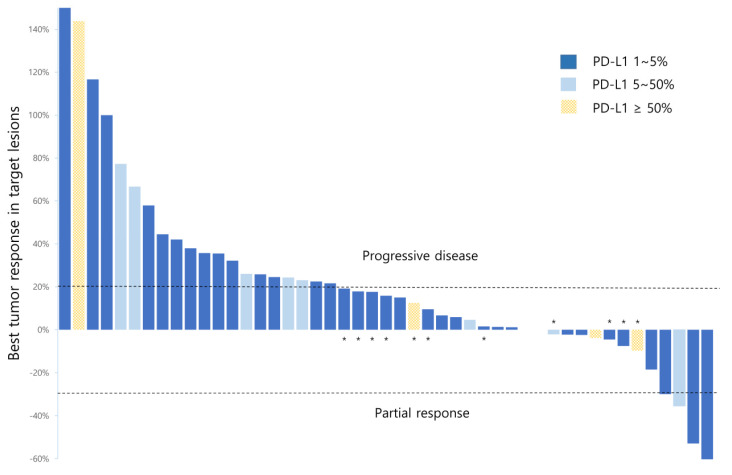
Tumor response and programmed death-ligand 1 (PD-L1) subgroups of evaluable patients (*n* = 42). The highest percentage changes in the size of tumor target lesions from the baseline, measured using Response Evaluation Criteria in Solid Tumors (RECIST) 1.1 for the 42 evaluable patients, are shown by PD-L1 subgroups. * Eleven patients with progressive disease based on RECIST 1.1 due to the unequivocal progression of non-target lesions despite <20% growth in target lesions.

**Table 1 jcm-09-01769-t001:** Baseline characteristics of patients.

Characteristics	
Age (years)	66 (43–83)
Sex (male)	29 (56.9%)
Performance status	
ECOG 0/1/2	24 (47.1%)/24 (47.1%)/3 (5.9%)
Location	
Intrahepatic/extrahepatic/gallbladder	12 (23.5%)/30 (58.8%)/9 (17.6%)
Stage	
Locally advanced/metastatic	6 (11.8%)/45 (88.2%) ^†^
Organ of metastasis	
Liver	24 (47.1%)
Intra-abdominal lymph node	24 (47.1%)
Peritoneal seeding	17 (33.3%)
Bone	4 (7.8%)
Lung	2 (3.9%)
Histological grading	
Well/moderate/poorly/unknown	3 (5.9%)/28 (54.9%)/14 (27.5%)/6 (11.8%)
PD-L1-positive (≥1%)	51 (100%)
1–5/5–50/≥50	35 (68.6%)/11 (21.6%)/5 (9.8%)
Laboratory results	
Hemoglobin (g/dL)	10.3 ± 1.8
Albumin (g/dL)	3.6 ± 0.5
AST (IU/L)	32 ± 19
ALT (IU/L)	35 ± 63
Total bilirubin (mg/dL)	0.72 ± 0.51
CA 19-9 (U/mL)	276.6 (1.1–24,253.0)
Clinical information	
Number of prior therapies (1/2/≥3)	33 (64.7%)/14 (27.5%)/4 (7.8%)
Cycles of pembrolizumab	3.0 (1–15)
≤3/>3	33 (64.7%)/27 (35.3%)
Combination treatment	1 (2.0%) ^‡^
Follow-up duration (months)	3.8 (0.6–18.4)

Patients, *n* = 51, ^†^ including 28 with post-operative recurrence. ^‡^ One patient who concurrently received palliative radiation therapy for cervical LN metastasis. Abbreviations: ECOG, Eastern Cooperative Oncology Group; PD-L1, programmed cell death 1-ligand 1; AST, aspartate aminotransferase; ALT, alanine aminotransferase; CA 19-9, carbohydrate antigen 19-9.

**Table 2 jcm-09-01769-t002:** Clinical outcomes.

Variable	
Partial response	5 (9.8%)
Stable disease	13 (25.5%)
Progression disease	33 (64.7%)
Disease control rate	35.3%
Time to response (months) ^†^	2.5 (2.1–8.3)
Progression-free survival (months)	2.1 (95% CI 1.7–2.4)
Overall survival (months)	6.9 (95% CI 5.4–8.3)

Patients, *n* = 51, ^†^ Evaluation in five patients with a partial response.

**Table 3 jcm-09-01769-t003:** Adverse events (AEs).

Variable	Any Grade	Grade 3/4
Adverse event	30 (58.8%)	4 (7.8%)
Non-hematologic		
Fatigue	11 (21.6%)	1 (2.0%)
AST/ALT elevation	8 (15.7%)	0
Nausea	5 (9.8%)	1 (2.0%)
Poor oral intake	3 (5.9%)	2 (3.9%)
Skin rash	3 (5.9%)	0
Diarrhea	2 (3.9%)	1 (2.0%)
Hypothyroidism	2 (3.9%)	0
Peripheral neuropathy	2 (3.9%)	0
Fever	1 (2.0%)	0
Facial edema	1 (1.8%)	0
Dizziness	1 (1.8%)	0
Dyspnea	1 (1.8%)	0
Drug-induced pneumonitis	2 (3.5%)	2 (3.9%)
Hematologic		
Anemia	3 (5.9%)	0
Thrombocytopenia	3 (5.9%)	0

Patients, *n* = 51; AST, aspartate aminotransferase; ALT, alanine aminotransferase.

**Table 4 jcm-09-01769-t004:** Predictors of progression after pembrolizumab treatment.

Variable	Univariate	Multivariate	
	*p*-Value	HR (95% CI)	*p*-Value
Sex (male)	NS		
Age (>65 years)	NS		
ECOG (>1)	NS		
Tumor location (intrahepatic CCA)	NS		
Stage (metastatic)	NS		
Histological grade (poorly differentiated) ^†^	NS		
PD-L1 group (1–5% vs. 5–50% vs. ≥50%)	NS		
Number or prior therapy (≥2)	0.018	1.917 (0.950–3.866)	0.069
Hemoglobin (<10 g/dL)	0.030	1.972 (0.966–4.026)	0.062
Albumin (<3.5 g/dL)	NS		
CA 19-9 (>500 U/mL)	0.013	1.751 (0.822–3.727)	0.146

Patients, *n* = 51; ^†^ excluding six cases of unknown histological grading. Abbreviations: HR, hazard ratio; CI, confidence interval; NS, not significant; CCA, cholangiocarcinoma; PD-L1, programmed cell death 1-ligand 1; CA 19-9, carbohydrate antigen 19-9.

**Table 5 jcm-09-01769-t005:** Reported data of pembrolizumab treatment in patients with gemcitabine/cisplatin-refractory biliary tract cancer (BTC).

Reference	Total *N*	ORR (*n*, %)	DCR (*n*, %)	PFS (months)	OS (months)	All AEs (*n*, %)	Grade 3/4 AEs (*n*, %)
Ott et al. (2019) [15,19]	23	4 (17%)	8 (34%)	1.8	6.2	15 (63%)	4 (17%)
Kang et al. (2019) [16]	40	4 (10.0%)	19 (47.5%)	1.5	4.3	8 (20.5%)	0 (0%)
Our data	51	5 (9.8%)	19 (35.2%)	2.1	6.9	30 (58.8%)	4 (7.8%)

Abbreviations: BTC, biliary tract cancer; ORR, objective response rate; DCR, disease control rate; PFS, progression-free survival; OS, overall survival; AE, adverse event.

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
