# Peer review of "Efficacy and Safety of Pembrolizumab for Gemcitabine/Cisplatin-Refractory Biliary Tract Cancer: A Multicenter Retrospective Study"

_jcm, 2020, doi:10.3390/jcm9061769_

Round 1
Reviewer 1 Report
In this work, authors described the effectiveness and safety of pembrolizumab in BTC patients refractory to Gem/Cis regimen. The reported a very modest result in terms of RR, PFS and OS. Adverse events are considerable, sometimes of grade 3-4.
Globally, this work does not provide new information compared to previous work on this topic. However, it confirmed the low effectiveness of this compound in such neoplasia.
Major revision
The number of patients enrolled in the study is low and there are different histotypes involved in the study. Recently, literature data suggested that they should be considered as different entities. Are there any differences in terms of response according to the histotypes? and According to PD-L1 expression? Is it possible to have KM curves?
Why did you use three different methods of IHC? why did you test only 10 samples with all the methods? It should be interesting analyzing all the samples eith the same method, to better compare the results.
Do you know if there are available in literature cutoffs different than >1% of positivity for PD-L1?
You have mentioned MSI as another biomarker of pembrolizumb response; there is also Tumor mutational burden, pease cite it.
Did PD-L1 evaluate on primary tumors before any systemic treatment? or for recurrent patients, a new biopsy was done? it should be interesting to know if the Pd-L1 changed from primary to recurrence.
Is sufficient one cycle of pembrolizumab to test the effect?
you wrote" all patients received 200 mg of Pembrolizumab.... after 17-day interval before treatment". 17 days from Gem/Cis?
You stated that prior therapies and hemoglobin levels could be related to the response to pembrolizumab and to progression. Couldn't be that the reason of such association is that previous therapies negatively influenced the patient's quality of life ?
How is the PD-L1 expression of the 5 patients who had PR, SD?
Do you think about which are the possibile biomarkers of response to pembrolizumab?
Minor revision
English revision is necessary.
Please, add the methods for IHC
Please, add the name of hospitals included in the study
Please, add the name of ethic committee who approved the study.
Reviewer 2 Report
This is a very interesting survey with regard to potential efficacy and safety of pembrolizumab for gemcitabine/cisplatin-refractory biliary tract cancer. Therefore, the article should be accepted for publication in its current form under minor revisions.
- Grammatical errors should be corrected throughout the Text.
- Parallel evaluation with recent studies' results should be performed in the Discussion section.
- Final conclusion should be more concise.
Round 2
Reviewer 1 Report
I think that this work is well presented, even if not so innovative. However, it confirms the limited effectiveness of pembrolizumab therapy in BTC patients. There are some weak points, (i.e. number of patients enrolled, different IHC methodics used). Authors should add a sentence about the study limitations.
Thank you for the exhaustive response.
